# Species-Level Differences in Osmoprotectants and Antioxidants Contribute to Stress Tolerance of *Quercus robur* L., and *Q. cerris* L. Seedlings under Water Deficit and High Temperatures

**DOI:** 10.3390/plants11131744

**Published:** 2022-06-30

**Authors:** Marko Kebert, Vanja Vuksanović, Jacqueline Stefels, Mirjana Bojović, Rita Horák, Saša Kostić, Branislav Kovačević, Saša Orlović, Luisa Neri, Massimiliano Magli, Francesca Rapparini

**Affiliations:** 1Institute of Lowland Forestry and Environment, University of Novi Sad, Antona Čehova 13d, 21000 Novi Sad, Serbia; kebertm@uns.ac.rs (M.K.); sasa.kostic@uns.ac.rs (S.K.); branek@uns.ac.rs (B.K.); sasao@uns.ac.rs (S.O.); 2Faculty of Agriculture, University of Novi Sad, Trg Dositeja Obradovića 8, 21000 Novi Sad, Serbia; vanja.vuksanovic@polj.uns.ac.rs; 3Groningen Institute for Evolutionary Life Sciences, University of Groningen, P.O. Box 11103, 9700 CC Groningen, The Netherlands; j.stefels@rug.nl; 4Faculty of Ecological Agriculture, Educons University, Vojvode Putnika 87, 21208 Sremska Kamenica, Serbia; mirjana.bojovic@educons.edu.rs; 5Teacher Training Faculty in the Hungarian Language, University of Novi Sad, Subotica, Štrosmajerova 11, 24000 Subotica, Serbia; rita.horak@magister.uns.ac.rs; 6Institute of BioEconomy (IBE), Department of Bio-Agrifood Science (DiSBA), National Research Council (CNR), Via P. Gobetti 101, I-40129 Bologna, Italy; luisa.neri@ibe.cnr.it (L.N.); massimiliano.magli@ibe.cnr.it (M.M.)

**Keywords:** Fagaceae, osmolytes, antioxidant, phytohormones, trade-off mechanisms, stress marker, oxidative stress

## Abstract

The general aim of this work was to compare the leaf-level responses of different protective components to water deficit and high temperatures in *Quercus cerris* L. and *Quercus robur* L. Several biochemical components of the osmotic adjustment and antioxidant system were investigated together with changes in hormones. *Q. cerris* and *Q. robur* seedlings responded to water deficit and high temperatures by: (1) activating a different pattern of osmoregulation and antioxidant mechanisms depending on the species and on the nature of the stress; (2) upregulating the synthesis of a newly-explored osmoprotectant, dimethylsulphoniopropionate (DMSP); (3) trading-off between metabolites; and (4) modulating hormone levels. Under water deficit, *Q. cerris* had a higher antioxidant capacity compared to *Q. robur*, which showed a lower investment in the antioxidant system. In both species, exposure to high temperatures induced a strong osmoregulation capacity that appeared largely conferred by DMSP in *Q. cerris* and by glycine betaine in *Q. robur*. Collectively, the more stress-responsive compounds in each species were those present at a significant basal level in non-stress conditions. Our results were discussed in terms of pre-adaptation and stress-induced metabolic patterns as related to species-specific stress tolerance features.

## 1. Introduction

Increasing air temperature and frequent summer drought events [1] will affect not only the health status, vitality, morphological, and physiological traits of different woody species but also their biochemical traits and even xeric distributional limits [2].

The prime response of plants to environmental constraints such as drought, high light, salinity, heavy metals or extreme temperatures is a reduction in net CO_2_ assimilation due to stomatal, mesophyll, and biochemical limitations [3,4,5,6]. This in turn results in an excess of light energy absorbed by chloroplasts relative to the capacity for photosynthesis and a consequent increase in the formation of reactive oxygen species (ROS, e.g., singlet oxygen, superoxide anion, hydrogen peroxide, and hydroxyl radicals [7,8,9].

Low amounts of ROS may have an important role in stress-signaling pathways, while high amounts of ROS detrimentally affect all biomolecules including lipids, proteins, DNA, and RNA, leading to oxidative stress and even to a programmed cell death [10].

Oxidative stress occurs when the production of excess ROS is not counterbalanced by the antioxidant defense system, so that unquenched ROS remains sustained to cause further reactions and oxidize biomolecules [10].

To counteract oxidative stress, plants activate a complex antioxidant network of enzymes (e.g., superoxide dismutases, ascorbate peroxidases, and catalases) and non-enzymatic compounds (e.g., ascorbate, glutathione, flavonoids, carotenoids, and phenolics) to defend plant cells [7,11] by controlling the production/scavenging of ROS. In addition, antioxidants can not only directly quench ROS activity, but they can also play an indirect role such as hormone-mediated signaling, upregulating primary defense genes, and activating secondary defense genes [12].

One central non-enzymatic component of the antioxidant system of plants is glutathione (GSH: γ-glutamyl-cysteinyl-glycine), the most abundant low molecular weight thiol which enhances plant tolerance to different abiotic stresses [13,14,15] acting either as a chemical antioxidant, a substrate of antioxidative enzymes, or as a substance for biotransformation of xenobiotics through the conjugation process [16]. The enhancement of responses of glutathione levels, glutathione turnover, and redox state in woody species under stress conditions results in an increased antioxidative capacity depending on plant species [13,17]. Among other potential stress markers in plants, flavonoids seem to be very efficient in drought and heat alleviation since many of them exhibit antioxidant properties against ROS [15,18]. Another important group of phenolic compounds are the condensed tannins (CT), particularly abundant in different organs of woody plants [19]. Quantity, composition, localization, and extent of polymerization of tannins are highly climate dependent [20]; however, there is still a lack of specific investigations on the impact of high temperatures and drought on their accumulation in oak species under controlled conditions.

When water is limited or temperatures are elevated, plants increase the production of osmotically active substances that have been associated with drought- and thermo-tolerance [15,21]. The enhanced accumulation of proline is regarded as protective response of plant metabolism against these stresses in many species [5,22]. Although glycine betaine (GB) is the most studied quaternary ammonium compound with defensive functions [23], however, there is insufficient knowledge about its production in woody tree species. Unlike the abovementioned N-containing osmolytes, little consideration has been given to the protective role of the tertiary sulphonium compound dimethylsulphoniopropionate (DMSP) in plant tolerance to abiotic stress [24], especially in woody species. The role of DMSP in plants was related to osmotic control and adjustments [25,26] and to antioxidant properties [27]. Although DMSP has recently been detected and quantified in leaves of 15 woody plant species, including *Quercus* (Kebert et al., unpublished data), its response to drought and heat stress in woody species was never examined.

To achieve a more complete understanding of the defense response to water deficit and high temperatures in the studied oak species, it is crucial to examine changes in the key phytohormones indole-3-acetic acid (IAA) and abscisic acid (ABA). They, indeed, play a pivotal role as developmental regulators under optimal growth conditions and in photoprotection of the photosynthetic apparatus under various stress conditions [28].

Under water limiting conditions, ABA is an important molecule in conveying the signals about water deficit from the soil to the roots and, in turn, from root to shoot, as it was proven that there is a special correlation between increased xylem sap ABA amounts and reduced stomatal conductance during water deficit [6,28].

The interaction between phytohormones and the components of the plant protection systems have been shown when plants are exposed to water deficit (i.e., ABA and xanthophyll cycle) [28]. Stress responsiveness of these hormones varies greatly depending on species/genotype [29,30], and their metabolic adjustments could reveal different species sensitivity and tolerance to stress conditions as previously shown in oaks, including *Q. cerris* [31]. In addition, potential cross-talk among phytohormones and proline has been suggested to occur in oaks [31].

Variation in the stress response of protective compounds is associated with differences in physiological plant performance and survival; however, the identification of species-specific metabolic sensitivities may provide early biochemical markers such as osmotic adjustments or antioxidant protection indicators that are informative for woody species tolerance to stress [32,33]. *Quercus* genus (*Fagaceae* family) includes more than 400 deciduous, evergreen, and shrub species [34]. Their responses to water deficit and heat stress are highly variable [35], and their interspecific variation in capacity to cope with these climate changes has not been elucidated [36]. Among the economically and ecologically most important deciduous oak tree species in Europe, the Turkey oak *Q. cerris* and the pedunculate oak *Q. robur* show similar ecological growing conditions but different leaf functional traits of protective systems (i.e., antioxidative and ormoregulative; Table 1). *Q. cerris* belongs to the group of the so-called ‘nemoro-Mediterranean oaks’ which occupy a xeric habitat [37]. The distributional range of this oak species extends from southern Europe to Asia Minor [38,39], and it is particularly present in the Balkan and Italian Peninsulas [36]. *Q. robur* is more widely distributed in Europe under a temperate-nemoral climate [39] (www.euforgen.org (accessed on 29 May 2022)). *Q. cerris* is known to be more drought tolerant than *Q. robur*, and recent studies evidence that both species are climate-sensitive as shown by the radial growth and stable carbon isotope records [40]. In particular, as predicted by climate scenarios, pedunculate oak is the most endangered *Quercus* species in the Balkan region, more than other *Quercus* species, including Turkey oak [41].

Although *Q. cerris* and *Q. robur* have developed effective protective mechanisms at both physiological and biochemical levels to counteract drought and heat stress [36], species-specific strategies involving antioxidant and osmoregulation systems have not yet been well elucidated, especially at the seedling stage and under high temperature stress (Table 1). In particular, the non-enzymatic antioxidants of *Q. robur* young plants were poorly investigated under both limiting soil water [42,43,44,45,46] and air warming conditions [44,45], generally only through the measurements of the ascorbate/glutathione system. Similarly, the osmotic adjustment of this oak species in response to both stresses has not been studied in depth [44,45]. The protective response of young plants of *Q. Cerris* has been investigated only under drought conditions and by examining only the accumulation of carotenoids among the non-enzymatic antioxidants and of proline among the compatible solutes [46].

When investigating the osmotic adjustment and production of antioxidants, however, the stress response of the protective components might vary from species to species or depend on the nature of the stress [32,33]. Generally, tolerant species are characterized by higher levels of protective metabolites, such as antioxidants and compatible solutes, under non-stress conditions and/or accumulate them in large amounts when stress occurs [47]. Therefore, the characterization of constitutive levels of protective compounds under optimal growth conditions may provide key information to assess the functional changes of these metabolites under stress conditions. In addition, the magnitude of stress-induced variations can reveal differences among genotypes, as species-dependent responses of protective compounds generally do not rely on the accumulation of a specific compound [47].

Differences in stress responsiveness of various defense metabolic traits may also be driven by the functional relationship (i.e., overlapping and complementary roles) between the individual metabolites contributing to the antioxidative and/or osmoprotective capacity of the plants [48,49]. Likewise, competition for mutual precursors between the synthesis of different compounds (i.e., proline vs. GSH or carotenoids vs. isoprenoids) may affect the endogenous levels of protective metabolites under stress conditions [17,50]. Therefore, the increase in investment in some components of the protection system can come at a cost to investments in the biosynthesis of other biochemical components if limiting resources are occurring, as during unfavorable conditions [51].

Therefore, the general aim of this work was to characterize the leaf-level capacity of antioxidant production and/or osmotic adjustments of *Q. cerris* and *Q. robur* in response to soil water deficit and increasing air temperatures through the measurements of a suite of different physiochemical parameters, including a newly-explored protective compound with a multifunctional role (i.e., DMSP). The examined seedlings from both species are expected to employ protective mechanisms to counteract oxidative stress under the applied stress conditions as the photosynthetic gas exchanges were reduced in response to the applied drought and heat stress on the same plants used for the present experiment [46,52,53]. The more specific objectives were to compare the constitutive and stress-induced changes in antioxidants, osmoprotectants, and phytohormones of the two oak species and to identify common and specific responses to drought and high temperatures.

The following hypotheses were tested:The drought and heat stresses result in specific changes in the production of both individual and/or pattern of osmoprotectants and antioxidants.Species-specific constitutive levels of antioxidants, osmoprotectants, and hormones predispose the studied oak species differently to unfavorable environmental conditions.The magnitude of the examined biochemical responses to both stresses depends on the species-specific features and on the nature of stress.Functional and metabolic relationships among the various metabolites contribute to the physiological performance and stress tolerance capacity of the investigated oak species.

By focusing on the plant capacity of osmotic adjustment and antioxidant production, our information on specific metabolic sensitivities of the investigated oak species to the applied abiotic stresses may provide early biochemical markers that are informative of woody species tolerance to stress [7,32]. Indeed, in forest management, both at the nursery and the field level, there is a need to identify measurable traits that predict susceptibility/resilience of species towards the ongoing climate change in light of the new concept of climate-smart-forestry-based monitoring systems [54].

## 2. Results

### 2.1. Effect of Drought and Heat Stress on Osmolyte and DMSP Accumulation

Proline (PRO) content was significantly affected by species and treatment (Appendix A; Two-way ANOVA Species, Treatment *p* < 0.001), with the treatment effect explaining about 75% of variation (based on sum of square values). Indeed, differences between species were observed only when plants were well-watered and under ambient temperatures (Controls-C; Figure 1A).

When oak seedlings experienced both stress conditions (water deficit-D, high temperatures-HS), a significant increase in PRO level occurred relative to control values, independent of species. In both *Quercus* species, the response of PRO to elevated temperatures was higher compared with that measured under water deficit, a pattern especially evident in *Q. cerris* (ca. 2.2-fold relative to control values in HS vs. 1.4-fold in D).

Although glycine betaine (GB) content in leaves was significantly affected by species and treatment (Appendix A; Two-way ANOVA Species, Treatment *p* < 0.001), in contrast to proline, the effect of species was the largest (based on *F* and Sum of square values). Both constitutive and stress-induced leaf levels of GB were significantly higher in *Q. robur* than in *Q. cerris* (Figure 1B). The interaction between the main effects was highly significant (Two-way ANOVA Species*Treatment *p* < 0.001). Indeed, the GB content increased in response to water scarcity, but the extent of the increase was species-specific: *Q. robur* showed a stronger response to water deficit (ca. 38% relative to control) in GB levels compared to *Q. cerris* (ca. 6% relative to control). Elevated temperatures induced a significant and similar increase in GB levels in both oak species (about 24% relative to control).

Similar to GB, dimethylsulphoniopropionate (DMSP) content in leaves was significantly affected by species and treatment (Appendix A; Two-way ANOVA Species, Treatment *p* < 0.001), and the effect of species was the largest (based on *F* and Sum of square values). Differences between species in leaf DMSP content were significantly evident both under control conditions and when seedlings experienced stress conditions, with leaf levels of *Q. cerris* significantly higher than those of *Q. robur* (Figure 1C). Examined oaks species showed increasing patterns of DMSP changes after exposure to both stresses, but the extent of the response depended on the species (Two-way ANOVA Species*Treatment *p* < 0.001). In *Q. cerris*, DMSP content markedly increased in response to water deficit and, especially, to high temperatures (about 2.4 and 4.2-fold relative to control values in D and HS, respectively). Differently, in *Q. robur*, the increase to both stress conditions compared to the control levels were significant but less dramatic (ca. < 1.7-fold relative to control values under both D and HS).

### 2.2. Effects of Water Deficit and High Temperatures on the Antioxidant Defense System

Leaf lipid peroxidation, measured as amount of malondialdehyde (MDA), was significantly and largely affected by treatment (Figure 2A; Appendix A; Two-way ANOVA Treatment *p* < 0.001) according to *F* values. The MDA content increased in response to water scarcity and high air temperatures, but within each species the extent of the increase was dependent on the nature of the stress. In *Q. robur*, the MDA levels were higher under elevated temperatures than under water deficit, while in *Q. cerris* they were similar.

Similar to MDA, the treatment factor mainly affected the levels of total non-protein thiols measured as reduced glutathione (GSH; Appendix A; Two-way ANOVA Treatment *p* < 0.001) that increased in response to water deficit and high temperatures (Figure 2B). The response to the applied treatments was dependent on species (Appendix A; Two-way ANOVA Species*Treatment *p* < 0.001). In particular, the response of GSH to high temperatures was two-fold higher in *Q. robur* compared to *Q. cerris* (47% vs. 16% relative to controls, respectively). Similar to MDA, after exposure to stresses, the GSH levels in the most responsive species (i.e., *Q. robur*) were higher under elevated temperature than under water deficit conditions.

Overall, the antioxidant capacity was significantly affected by treatment, and the interaction between treatment and species was significant (Appendix A; Two-way ANOVA Treatment *p* < 0.001 for ABTS, DPPH, FRAP, Species*Treatment *p* < 0.01 for ABTS, *p* < 0.001 for DPPH and FRAP). The increasing effect of water deficit on the measured parameters was significantly pronounced in *Q. cerris* (Figure 3; about 52% and 25% relative to controls for DPPH and FRAP), while in *Q. robur* this stress treatment either marginally affected or did not affect most of these parameters. Similar to water deficit response, the antioxidant capacity was responsive to high temperatures in *Q. cerris*, while it was not affected in *Q. robur*.

The total phenolic content (TPC) was significantly affected by species and treatment (Appendix A; Two-way ANOVA Treatment *p* < 0.001, species *p* < 0.01) as well by their interaction (Species*Treatment *p* < 0.001). Similar to the overall antioxidant capacity, only in *Q. cerris* was this component of the antioxidant defense system significantly responsive to water deficit and high temperatures, both conditions inducing an increase or a decrease compared to control plants, respectively (Figure 4A).

The main effect of species on total flavonoid content (TFC; Appendix A; Two-way ANOVA Species *p* < 0.001) was shown by *Q. robur* having a significantly higher constitutive content of these compounds than *Q. cerris* (Figure 4B). This species-specific difference was maintained under stress conditions. Indeed, in each species, TFC was not significantly altered by water scarcity or elevated temperatures, confirming the lower relevance of the treatment effect based on the *F* values of two-way ANOVA analysis.

The basal level of condensed tannins (CT) under control conditions was higher in *Q. cerris* than in *Q. robur* (Figure 4C). In both oak species, CT increased in response to water scarcity and high temperatures, but the extent of the increase depended on the species and the nature of the stress (Appendix A; Two-way ANOVA Species*Treatment *p* < 0.001). The increasing effect of water deficit on CT was significantly more pronounced in *Q. cerris* (3-fold relative to control) than in *Q. robur* (1.8-fold). After exposure to high temperatures, in *Q. cerris* there was again a more significant increase of CT than in *Q. robur*. In both oak species, water deficit-related changes were significantly stronger than those induced by high temperatures relative to controls.

### 2.3. Leaf Nitrogen and Sulfur Status in Response to Water Deficit and High Temperatures

Overall, the leaf nitrogen (N) and sulfur (S) content was affected by both stress conditions, but the effect depended on the species (Appendix A; Two-way ANOVA Species*Treatment *p* < 0.001 for N and *p* < 0.01 and S). In *Q. robur*, soil water deficit and elevated air temperatures induced a decrease of N and S content, while in *Q. cerris*, both stress conditions either slightly affected or did not affect the leaf content of N and S, respectively (Appendix A).

### 2.4. Changes in Leaf IAA and ABA Levels Induced by Soil Drying Conditions and High Temperatures

Under control conditions, both oak species were characterized by comparable leaf levels of indole-3-acetic acid (IAA; Figure 5A; ca. 382 ng g^−1^ DW). Water deficit induced a significant accumulation of IAA in *Q. cerris*, but not in *Q. robur*. Elevated temperatures did not affect IAA concentrations in both species.

Different from IAA, the leaf constitutive ABA levels were significantly higher in *Q. cerris* than in *Q. robur* (Figure 5B; 242 ng g^−1^ DW vs. 101 ng g^−1^ DW, respectively). The observed effects of stress treatment on ABA concentrations depended on the species (Appendix A; Two-way ANOVA Treatment, Species*Treatment *p* < 0.001). Indeed, in response to water deficit, ABA levels increased in *Q. robur* (ca. five-fold compared to controls), while they did not change in *Q. cerris*. By contrast, after exposure to high temperatures, ABA content slightly increased in *Q. robur* (*t*-test *p* < 0.05), while it was significantly reduced in *Q. cerris* (ca. 44% relative to controls).

### 2.5. PCA Analysis

To test if species shared a common or specific pattern of response to the imposed stress conditions, a principal component analysis (PCA) was run on the full data set (n = 90 cases/270 observations), and an acceptable solution was reached when two principal components described 72.0% of the total variance of the original data set (Figure 6).

As expected, component 1 (PC1) captured the highest variance within the data (41.8% of the observed variance) and mostly accounted for differences between species as evidenced by the greater effect of species on PC1 than that of the treatment according to *F* values of Two-way ANOVA (Appendix A). In particular, species-specific differences were evident across drought-stressed samples, with PC1 strongly discriminating the water-deficit-stressed samples of *Q. cerris* from those of *Q. robur*.

PC1 segregated species mainly according to a combination of antioxidant components and hormonal contribution. The response of *Q. cerris* to water deficit mainly corresponds to high loadings of the highly correlated antioxidant parameters (FRAP, ABTS, and DPPH), total phenolic (TPC), total condensed tannins (CT), and IAA content. The opposite position of GB to antioxidant systems is associated with *Q. robur* stressed plants, that are also characterized by high levels of TFC, N, and S and reveals that this species under water deficit conditions is marginally characterized by the activation of the antioxidant systems, while mainly relying on accumulation of the specific osmolyte (GB) or specific antioxidants (i.e., flavonoids).

Component 2 (PC2) was responsible for 30.2% of the observed variance and mostly reflected the effect of the treatment factor that was greater than that of species on PC2 compared to PC1 (*F* values of Two-way ANOVA; Appendix A). Indeed, PC2 scores clearly separate the response to water deficit from the one induced by high temperatures for both *Quercus* species. In both oak species, exposure to high temperatures appeared a relevant source of variance in the data, which indicates a significant change of metabolism under these stress conditions. The significant interaction term (Species*Treatment) highlighted that such a response pattern of PC2 was dependent on species-specific differences. Indeed, PC2 captured the effect of elevated temperatures in each species by clearly separating the heat-treated samples from their controls, with *Q. cerris* showing better performance compared to *Q. robur*. High scores in this component correspond to high loadings of oxidative stress markers (MDA, GSH) and of osmoprotectants (PRO, DMSP), indicating in both species a strong activation of osmoregulation under high temperatures.

### 2.6. Heat Map with Bi-Cluster Analysis

Hierarchical clustering heat map analysis showed differences between oak species and between the metabolites in response to water deficit and high temperatures (Figure 7).

The species-specific response to water scarcity was evidenced by the drought-treated samples of each oak species being clustered distant. Under this stress condition, the heat map data clearly indicated that some metabolic responses were specifically induced in each oak species. In particular, the typical response of *Q. robur* to water deficit involves the upper cluster in the metabolite hierarchy that contained GB and ABA. This species was also characterized by maintaining under this stress condition enriched flavonoid reserves that were comparable to those under pre-stressed conditions. The distinct response of *Q. cerris* to water deficit involved the lower metabolite cluster, including phenolics, tannins, and parameters indicative of antioxidant capacity (FRAP, ABTS, DPPH).

The clustering of samples showed that in both *Quercus* species the biochemical response to elevated temperatures differed from the correspondent responses to water deficit. In contrast to water deficit, the high-temperature-treated samples of the two oak species were very close and displayed a positive correlation with the metabolite cluster that includes oxidative stress markers and osmolytes, reflecting the above described PCA observations. Under high temperature conditions, slight differences between oak species were related to the magnitude of changes in proline and GSH, while a distinguished upregulation of specific osmoprotectants was evident (i.e., GB and DMSP in *Q. robur* and *Q. cerris*, respectively).

### 2.7. Correlation Analysis

Findings by PCA and heat map cluster analysis almost completely matched the relationships between metabolites suggested by the correlation coefficients (Figure 8). For example, a high significant and positive correlation was found among oxidative stress markers (GSH, MDA) and osmolytes (DMSP and PRO) with correlation coefficients ranging from 0.73 to 0.80. Similar elevated correlation coefficients (i.e., ranging from 0.72 up to 0.95) were significantly found (*p* < 0.001) among the antioxidant parameters (DPPH, ABTS) and the content of phenolics and tannins.

The observed negative correlations among total flavonoid content (TFC) and the amount of both PRO and DMSP is consistent with PCA results (i.e., opposite position of TFC to PRO and DMSP) and with the heat map data. Taken together, the observations further underline that an increase of osmolyte content under high temperatures is associated with a decrease in flavonoid content.

Interestingly, most of the parameters related to the antioxidant defense system (i.e., CT, TPC, ABTS, DPPH, FRAP) showed a significantly high and positive correlation with the IAA content, with correlation coefficients ranging from 0.69 to 0.89.

The parameters CT, TPC, and those of the antioxidant capacity (FRAP, DPPH, and ABTS) similarly covariate, showing that they are part of the same antioxidant system. High significant and positive correlations among DPPH, TEAC, and FRAP indicate that these parameters use the same mechanism of antioxidant action since practically all of them measure antioxidant capacity that employs an electron transfer (ET) mechanism, which is thoroughly explained by different authors [55,56]. These assays use the assumption that antioxidant capacity is equal to the reducing capacity of extracts [55].

## 3. Discussion

*Quercus* species have been widely studied in response to environmental stress conditions [40,57]. However, the discrimination of antioxidative and osmoregulation traits among species is still not well established, especially in young trees and under air warming conditions (Table 1).

Our findings show that Turkey oak (*Q. cerris*) and pedunculate oak (*Q. robur*) seedlings responded to the applied soil water deficit and high air temperature conditions by (Figure 9): (1) increasing differently the levels of several components of their osmoregulation and antioxidative defense system depending on the species and the nature of the stress; (2) upregulating the synthesis of the newly-explored metabolite dimethylsulphoniopropionate (DMSP); (3) trading-off between multifaceted components of the protection system; and (4) modulating the synthesis of stress-related (ABA) and developmental-related (IAA) hormones.

The occurrence of oxidative stress in leaves of both species under both soil water deficit and high air temperature conditions (i.e., increased membrane lipid peroxidation), is consistent with similar results in *Quercus* [58], including *Q. cerris* and *Q. robur* (Table 1) [42,59,60]. However, the low level of lipid peroxidation, together with the increasing response pattern of mostly biochemical parameters to both stress conditions, suggest an acclimation to oxidative stress for the examined species [61].

Tolerance to the imposed stress conditions was evident in previous measurements of the physiological status on the same plants used for the present experiment [46,52,53]. Although under soil water deficit, a reduced leaf CO_2_ assimilation and stomatal conductance was detected, the photosynthetic rates were still rather high (ca. 7–9 µmol m^−2^ s^−1^ in *Q. cerris* and ca. 9–16 µmol m^−2^ s^−1^ in *Q. robur*) [46,53,62,63]. Similarly, thermotolerance of the investigated species was revealed by the maintenance of significant photosynthetic functionality, together with the lack of irreversible damage to the photosynthetic apparatus (i.e., photoinhibition of the PSII) even at the highest temperature (47 °C; [52,64]).

### 3.1. Species-Specific Accumulation of Osmotically Active Substances Is Induced Differently by Drought and Elevated Temperatures

Under both water deficit and high temperatures, the significant positive covariation between the levels of osmoprotectant proline and DMSP and those of the oxidative stress markers MDA and GSH indicated that an active osmoregulation took place to limit oxidative damage.

When water is limited and a decrease of stomatal conductance occurs together with an increase in diffusive resistance to CO_2_, plants can rely on mechanisms of osmotic adjustment as a strategy for maintaining turgor at low leaf water potentials and reducing osmotic potential through the accumulation of compatible solutes in plant leaves [65]. After a drought period similar to that applied in the present study (i.e., within 15 days), acclimation of oak seedlings generally involves osmotic adjustment [66]. The observed increased content of proline is a widely prevalent response to drought due to its multiple functions [22], as previously reported in young trees of *Q. cerris* [59,60,67] (Table 1; Cotrozzi et al., 2016; 2017) and in seedlings of *Q. robur* under soil water regimes (soil moisture ranging between 15- 25%) similar to the present study [43,68] (Table 1; Spieß et al., 2012).

Our findings show that the expected water deficit induction of other osmolytes is characterized by a different extent of the response depending on species, as particularly evidenced by PCA and bi-cluster analysis. In *Q. robur*, glycine betaine showed higher constitutive levels and was more drought-responsive than in *Q. cerris*, suggesting a relevant function of this N-containing compound in building drought tolerance of the former species. On the other hand, osmoregulation of *Q. cerris* appears largely conferred by the S-analog of glycine betaine, namely dimethylsulphoniopropionate (DMSP), which shows higher basal pools and water deficit-induced increases of this osmolyte compared to *Q. robur*. Among the metabolically compatible osmolytes, the role of DMSP is revealed for the first time in the *Quercus* species. This crucial component of sulfur metabolism has scarcely been investigated, especially in woody species [69,70]. The detected DMSP levels occurred at a concentration range (nmol g^−1^ DW) similar to that previously reported in leaves of other species [70,71,72]. The accumulation of DMSP in response to soil water deficit, especially evident in *Q. cerris*, is consistent with previous findings of drought-induced DMSP production in herbaceous species [70,72]. DMSP has been proposed as having a multifunctional role in response to reduced water availability, acting not only as osmoprotectant, but also as antioxidant and overflow for excess energy, thus protecting the photosynthetic machinery from stress injury [69,70].

When oak seedlings were exposed to high temperatures, a strong osmoregulation capacity was mostly involved and enhanced, independent of species, as clearly evidenced by PCA and bi-cluster analysis. Under these stress conditions, the higher oxidative pressure (i.e., elevated MDA and GSH values) was concomitant to a higher accumulation of osmolytes, in particular of proline, compared to the correspondent changes under soil water deficit. This amino acid acts as an effective ROS scavenger in the protection against denaturation and in the stabilization of membrane and subcellular structures [73,74]. Despite the general pronounced induction of osmoregulation in both species when exposed to high temperatures, heat stress impact each species differently as clearly shown by heat map analysis. Similar to the impact of water deficit on osmolyte levels, under high temperatures the stress tolerance of the examined *Quercus* species appears to rely on the major contribution of specific molecules. In *Q. robur*, the pronounced heat-induced oxidative stress is counteracted by sustained and elevated pools of glycine betaine, while in *Q. cerris* the marked accumulation of DMSP appears to play a major role. The protective role of glycine betaine is well recognized under temperature stress [75], while the function of DMSP in thermal tolerance is still unknown, its role being understood only in response to osmotic stresses (i.e., water scarcity or salinity; [72,76,77]).

Taken together, it is worth noting that within each species the magnitude of the stress-induced responses of osmolytes is dependent on the nature of stress. In *Q. cerris*, the specific DMSP changes to high air temperatures were particularly higher than those to water deficit. On the other hand, in *Q. robur* the typical glycine betaine response to soil water deficit was stronger than under the high temperature treatment.

When seedlings were exposed to high temperatures, the marked increase of N- and S-containing osmolytes (i.e., proline and DMSP) was associated with decreased (*Q. robur*) or almost unchanged (*Q. cerris*) levels of leaf inorganic N and S. These findings suggest that in *Q. robur* the enhanced formation of osmolytes may be at the expense of leaf structural N and S, as previously found in oaks [45,78], while in *Q. cerris*, changes in whole plant distribution of inorganic N and S may occur.

### 3.2. Foliar Antioxidant Defense Systems Are Activated by Drought but Less by Heat

In both oak species, the concomitant and highly correlated increase of foliar MDA and total non-protein thiols (i.e., GSH) under the applied stress conditions can be considered indirect evidence of plant activation of antioxidant responses as previously found in these species (Table 1). However, the induction of individual constituents of the antioxidant system can differ depending on species [79,80] as previously shown in oaks [36,44,45,81] also at molecular level [82].

In *Q. cerris* seedlings, the enhanced antioxidant capacity of leaves in response to soil water deficit can be attributed to the specific accumulation of phenolic compounds and to the higher responsiveness of tannins compared to *Q. robur*. Indeed, these correlated antioxidant components support the occurrence of a concerted defense action involving antioxidant function when water availability is scarce, while highlighting a typical response pattern of *Q. cerris*. Likewise, the slight decrease or no significant changes in the antioxidant capacities of *Q. robur* leaves under soil water deficit was associated with the lack of phenolic accumulation. The less efficient antioxidant defense, in terms of both radical scavenging activity and concentrations of some antioxidants, of *Q. robur* agrees with previous findings obtained on the same oak species and under similar conditions of moderate and relatively short drought stress [43].

When plants were exposed to high air temperatures, a species-specific response of tannins was evident and similar to that observed under soil water deficit, with *Q. cerris* exhibiting a higher responsiveness of these antioxidants to both stress conditions than *Q. robur*. Drought- and heat-induced increasing patterns of foliar tannins have been found in other woody species under stress conditions [83], and our findings confirm that leaf content of tannins is highly species-dependent [84]. Similar to the detected changes of specific osmolytes, we cannot exclude that composition or single metabolites belonging to the classes of phenolics, tannins, or flavonoids respond differently to the imposed stress conditions, depending on species. In addition, we are aware that the analysis of the enzymatic component would provide a more robust understanding of the differential role of the antioxidant machinery in stress tolerance of both *Quercus* species to each specific environmental stressor.

### 3.3. Tolerance to Soil Water Deficit and High Temperature May Rely on Compensation Mechanisms

Collectively, our findings suggest that under the applied environmental stress conditions a potential and slight trade-off or compensation mechanism may have occurred between the measured components of the antioxidant defense system [17,85]. Protective compounds, such as proline, glycine betaine, and DMSP, play a major role as osmoprotectants but also may act as antioxidants. Therefore, the observed enhancement of proline, glycine betaine, and DMSP in response to both stress conditions could make the activation of other components of the antioxidant defense system less necessary. For example, in both oak species under elevated temperatures, results of PCA analysis together with the general negative correlation between proline and flavonoids may indicate a functional trade-off between these components of the same protective systems. This mechanism may also explain the lack of activation of an antioxidant capacity in *Q. robur* when the investment on other antioxidant components such as flavonoids is sustained. Taken together, these results are also in agreement with previous results suggesting similar compensation mechanisms in other *Quercus* species [86] including *Q. robur* [44].

The marked increase of DMSP in response to the imposed stress conditions appears interesting as this sulfur compound has a metabolic pathway and a functional relationship interacting with that of another protective compound, namely isoprene [70,87], which acts as an antioxidant under drought and thermal stress in several woody species [88] including *Quercus* [89]. Although we did not measure isoprene emission rates of the examined seedlings, the well-established different isoprene emission ability of the two oak species (i.e., *Q. robur* is a high isoprene emitter, *Q. cerris* is a low isoprene emitter; [90]) could further explain their distinct activation of the examined protection systems. In particular, a metabolic trade-off between DMSP and isoprene synthesis may in part justify the high production of DMSP measured in the low isoprene emitter *Quercus cerris*. On the other hand, the minor response of DMSP under high temperatures in the high isoprene emitter *Q. robur* could be partially explained by a trade-off mechanism produced between DMSP and isoprene not only in term of resource allocation for their respective synthesis, but also as a consequence of the antioxidant function of isoprene that in turn can have a potential negative consequence on other traits [51]. Indeed, it has been suggested that the isoprene ability to protect the photosynthetic apparatus [91] can decrease the oxidative load on other leaf protective components [92]. The strong isoprene emission rates of *Q. robur* leaves under high temperatures [93] further support the possibility of the proposed compensation mechanism. Likewise, the lack of induction of leaf antioxidants when *Q. robur* plants were exposed to water deficit is consistent with a reduced requirement for an increased response of other antioxidants, as previously reported in other isoprene-emitting species [92,94] including oaks [89]. However, the influence of isoprene emission capacity on the activation of antioxidants is still controversial [91,95], and further experiments should be designed specifically to explore this hypothesis.

### 3.4. Drought and Heat Stress Impact the Hormone Balance Differently

Soil water deficit triggered the increase in leaf IAA concentrations in both species but particularly in *Q. cerris*. The strong and significant positive correlations between the extent of IAA response and that of several antioxidants was particularly evident in *Q. cerris* while almost lacking in *Q. robur*. These results suggest that in Turkey oak an enhanced leaf IAA content can contribute to an improved stress tolerance in accordance with findings reported in other woody species under drought conditions [96]. Auxins are recognized to have an impact on photosynthesis with an enhancement of drought tolerance [97].

On the other hand, the increased leaf ABA concentrations in *Q. robur* are an expected plant response to soil water deficit [98,99] including woody species [99,100]. Leaf ABA accumulation is frequently linked with drought tolerance as this hormone plays a key regulatory function in controlling stomatal aperture, in regulating stress-related genes, and in stress signaling [98,99]. The lack of ABA response to drought observed in *Q. cerris* could reflect the moderate stomatal control of transpiration of this anisohydric tree species when subjected to soil water deficit [101]. On the other hand, the high leaf ABA constitutive levels observed in *Q. cerris* may be an important adaptive feature for water conservation in drought-tolerant trees [102].

High temperature conditions did not influence the endogenous leaf concentrations of IAA in both oak species. The role of IAA in heat tolerance is still controversial. Although it is generally assumed that heat results in an increase in IAA content [103], the lack of responses of IAA levels to high temperatures in our study might be explained by the dynamic nature of IAA. Indeed, in model plants, growth temperature can alter IAA turnover rates and biosynthetic route without a correlated change in its absolute levels [65].

The ABA decrease observed in *Q. cerris* in response to high temperatures might result from substrate competition for common precursors between the biosynthesis of this phytohormone and photosynthetic pigments as previously found under stress conditions [50,104]. After exposure to high temperatures, significant changes in the antioxidant carotenoids were found on the same plants of the present study [52], reflecting a typical response to oxidative stress in several species [85,89] including *Quercus* [42,59,81,105]. Additional explanations of the lack of ABA induction under the applied temperature stress conditions can also rely on the increased stomatal conductance of the same seedlings as a possible consequence of the unlimited water supply [52]. Overall, we cannot exclude that the hormonal balance between the stress-related ABA and the developmental-related IAA, rather than the activities of each single hormone, might contribute coordinately to the tolerance to drought and heat [106].

## 4. Materials and Methods

### 4.1. Plant Material and Experimental Design

Experiments were conducted using seedlings of pedunculate oak (*Quercus robur*) and Turkey oak (*Quercus cerris*) as at that developmental stage plants are more sensitive to environmental stresses than adult trees. Therefore, seedlings are more likely to express differences due to water deficit or high temperatures stress, providing relevant information on the tolerance characteristics of the investigated species. Acorns of both oak species were collected from a natural oak population at Morović, which is the largest natural pedunculate oak forest in Serbia. Seeds were germinated in vermiculite in a climate chamber at 25 °C, 80% humidity. Seedlings were transferred to 5 L Micherlich pots filled with loamy fluvisol soil. Soil properties were as follows: pH 8.1 in H_2_O, (7.6 in KCl), 21.9 mg g^−1^ humus, 1.32 mg g^−1^ nitrogen, 10.52 mg g^−1^ Mg, 12.68 mg g ^−1^ Ca, 7.5 mg g^−1^ K_2_O, and 124.5 mg g^−1^ CaCO_3_. Plants were grown for 3 months (April to June) in semi-controlled conditions (ambient temperature between 25–30 °C) at the greenhouse of the Faculty of Science, Department for Biology and Ecology at the University of Novi Sad, Novi Sad (Serbia). Plants were watered to the soil’s maximum water capacity by replacing the amount of water transpired every day until the onset of the experiment.

Soil water content (SWC) was determined daily by drying soil samples (1 g) at 105 °C for approximately 48 h until constant mass was reached and expressed as volume percentages (% vol) according to the formula:$$\mathrm{SWC}\;[\%]=\frac{\mathrm{mass}\;\mathrm{of}\;\mathrm{water}}{\mathrm{mass}\;\mathrm{of}\;\mathrm{dry}\;\mathrm{soil}}\times 100\%$$

After determining SWC, the Mitscherlich pots were filled with soil. The plants were grown for 90 days under optimal water supply conditions (they were filled every other day, with SWC ranging from 29 to 38% vol). The day before the start of the treatment, all plants in the Mitscherlich pots were irrigated to the maximum, and the mass of each pot was measured at the maximum substrate moisture. Three-month-old plants (each with 5 to 10 leaves, without side branches) were subjected to treatments.

The experiment consisted of four treatments with a randomized block design as follows:(1)Control (C): plants were well-watered daily to maintain SWC in the range of 32–38% and were grown at ambient temperature (daytime temperature: 25–28 °C; nighttime: 10–14 °C);(2)Drought (D): plants were water stressed by withholding water for 12 days until the SWC reached values of ca. 9–11%. The duration of the drought treatment was chosen to induce an almost natural, reversible drought stress, thus allowing the plant enough time to acclimate and to recover [52,63];(3)Heat stress (HS): plants were exposed to daily air temperatures ranging between 33–47 °C (night temperatures were about 25–30 °C), using a small enclosure chamber in the greenhouse for 6 days [64]. During the heat treatment, plants were regularly watered to maintain the soil’s maximum water capacity, and the daily air temperature was continuously monitored with thermo-sensors.

### 4.2. Measurements of Osmolytes’ Accumulation:

Proline (PRO) concentration was estimated spectrophotometrically at 520 nm after the reaction of proline with ninhydrin reagent by using a rapid colorimetric method described by [107]. Glycine-betaine (GB), as the predominant quaternary ammonium compound (QAC), was quantified using the precipitation method of QAC-periodide complexes in an acid medium [108]. Dimethylsulphoniopropionate (DMSP) content was quantified by measuring the volatile compound dimethyl sulfide (DMS) released after basic hydrolysis from leaf material and swept out into a proton transfer reaction mass spectrometer (PTR-MS, Ionicon, GmbH, Innsbruck, Austria) as thoroughly described by Stefels [109]. All osmolyte data were calculated on a DW basis.

### 4.3. Assays of Antioxidant Defense Systems

Fully developed leaves of *Q. robur* and *Q. cerris* were sampled from each treatment, frozen, and grinded in liquid nitrogen and later lyophilized in a freeze dryer at −80 °C prior to analysis. For the different chemical analyses, either freeze-dried material was used directly, or extracts in ethanol or phosphate-buffered saline (PBS; 0.1 M KH_2_PO_4_, KOH, pH = 7) were prepared. Ethanolic extracts were prepared in 2 mL test tubes by mixing around 0.1 g of powdered freeze-dried leaf material with 2 mL of ethanol (96%). Samples were vigorously vortexed and then centrifuged for 30 min at 13,200 rpm at 4 °C. The supernatant was used for the determination of total phenolic and total flavonoid contents, as well as for the quantification of the antioxidant activity of plant extracts by three different methods (see below). Extracts obtained by mixing 0.1 g of plant material with 2 mL of PBS buffer were used for the determination of total non-protein thiol and malondialdehyde (MDA).

Therefore, to investigate the antioxidant capacity of the selected plant species, the following non-enzymatic biochemical parameters were measured. The response of the antioxidant system was investigated through the estimation of several components of the oxidative stress: oxidative damage and antioxidant defenses (i.e., total antioxidant capacity against radical attack and endogenous levels of non-enzymatic antioxidants):(1)Lipid peroxidation was measured as malondialdehyde (MDA) equivalent which is the secondary end-product of the oxidation of polyunsaturated fatty acids. Determination of MDA was carried out using the acid-catalyzed complexation reaction between MDA and thiobarbituric acid [110]; results are expressed as nmol MDA equivalents on a DW basis.(2)Total non-protein thiol compounds were measured according to a modified colorimetric assay based on measuring the absorbance of yellow Ellman’s reagent (5,5′-dithiobis-(2-nitrobenyoic acid; DTNB) reduced by sulfhydryl compounds at 413 nm. After construction of the calibration curve where we used reduced glutathione (GSH) as standard, total non-protein thiol compounds were expressed as GSH equivalents on a DW basis [111].(3)Trolox^®^ Equivalent Antioxidant Capacity (TEAC) was estimated with the 2,2′-azinobis-(3-ethylbenzothiozoline-6-sulfonic acid) (ABTS) assay based on the capability of the ethanolic extract to scavenge ABTS radicals [112]. Data were expressed as TEAC on a DW basis.(4)The 2,2-di-phenyl-1-picrylhydrazyl (DPPH) radical scavenging assay was also applied to measure the antioxidant activity level of ethanolic extracts expressed as TEAC on a DW basis [113].(5)The Ferric Reducing Antioxidant Power (FRAP) assay was performed to estimate the ability of the plant extract to reduce the ferric 2, 4, 6-tripyridyl-S-triazine complex [Fe^3+^-(TPTZ)_2_]^3−^ to the intensively blue-colored ferrous complex (Fe^2+^-(TPTZ)_2_]^2−^ in acidic medium [114]. Data are expressed as TEAC on a DW basis.(6)Total phenolic content (TPC) was measured by the Folin–Ciocalteu method [115]. Data are expressed as mg of Gallic Acid Equivalents (GAE) on a DW basis.(7)Total flavonoid content (TFC) was measured by the aluminum chloride colorimetric method [116]. Data are expressed as mg of Quercetin Equivalent (QE) on a DW basis.(8)Condensed tannins (CT) content was determined from methanolic extracts by using the butanol-HCl-Fe (III) method [117]. Data were expressed as leucocyanidin equivalents (LE) on a DW basis.

### 4.4. Elemental Analysis of Inorganic Nitrogen and Sulfur

The total content of inorganic nitrogen (N) and sulfur (S) were determined from freeze-dried and powdered oak leaf samples (5–10 mg) with a CHNS analyzer, model Elemental VARIO EL III [118].

### 4.5. Determination of Leaf Indol-3-Acetic Acid (IAA) and Abscisic Acid (ABA)

A highly reproducible extraction and purification procedure developed for leaf material of woody species was adapted from [4]. Phytohormones (IAA and ABA) were extracted from freeze-dried leaf samples with an extraction mixture (65:35, isopropanol: 0.2 M imidazole buffer, *v*/*v*, pH 7.0) and spiked with isotopically labeled internal standards [^13^C_6_] IAA and [^2^H_4_] ABA (OlChemIm Ltd., Olomouc, Czech Republic). Extracts were purified by using a 3 mL bed volume Quaternary Amine (Strata SAX, Phenomenex, CA, USA) solid phase extraction (SPE) columns, and later, the aqueous phase was additionally purified by using 3 mL volume C_−18_ (STRATA C_−18_, Phenomenex, CA, USA) SPE columns, whereas IAA and ABA were finally eluted with acetonitrile, evaporated to dryness, and re-suspended in 100 μL methanol. Derivatization of methanolic extracts was carried out with freshly prepared diazomethane [119]. The methylated product was dried under a stream of N_2_ and resuspended in ethyl acetate for analysis by gas chromatography coupled with mass spectrometry (GC-MS, model 7890A-5975C, Agilent Technologies, Santa Clara, CA, USA) in split-less mode [120,121]. Methyl esters of plant hormones were separated on an HP1 capillary column (60 m × 0.25 mm, 0.25 μm film thickness, Agilent, Santa Clara, CA, USA) using He as the carrier gas and temperature program as it was described in [4]. IAA and ABA were determined by using the Selected Ion Monitoring (SIM) mode, and final concentrations were calculated by calculating the ratios between corresponding peak and internal standard peak, peak area of m/z 130/136 and m/z 189/195 for IAA, and m/z 190/194 and 162/166 for ABA, according to the principles of isotope dilution [65,121].

### 4.6. Statistical Analysis

All statistical analyses were run with R studio software (R Studio, Boston, MA, USA). Prior to analysis, Shapiro–Wilk and Levene’s tests were performed to evaluate normality and homogeneity of variance of all response variables. Log (x) or Log(x + 1) transformations were applied to some data (ABTS, FRAP, DPPH) to account for violation of assumptions of normal distribution and homogeneity of variance. To determine the overall significances between groups, two-way factorial analysis of variance (two-way ANOVA) was used with species and treatments as factors. The statistical significance of the differences in the means were determined: (i) across treatments within each species, and (ii) across species within each treatment using Tukey’s honest significant difference (HSD) post hoc test for multiple comparisons when the one-way analysis of variance (ANOVA) indicated significant differences (*p* ≤ 0.05). Pearson’s correlation analysis was conducted to explore the associations between the examined parameters. A principal component analysis (PCA) was used to identify patterns of variations of the examined parameters across species and treatments. Before performing the PCA, the values of each parameter were standardized. A two-way ANOVA was also performed on the scores of the PC explaining the highest proportion of variance (i.e., PC1 and PC2) to test the effect of the factor species and treatment and their interaction on the extracted collective variables. Grouping of treatments and measured traits was examined by hierarchical cluster analysis. Expander 7 was used to create a heat map with bi-cluster analysis based on corresponding Pearson’s coefficient. All graphical representations were carried out using R package. All statistical tests were based on three independent repetitions (individual seedlings) for each treatment and species.

## 5. Conclusions

In summary, when seedlings of pedunculate oak (*Q. robur*) and Turkey oak (*Quercus cerris*) experience soil water deficit and high air temperatures they activate a different pattern of osmoregulation and antioxidant mechanisms to counteract and minimize oxidative stress depending on the species and on the nature of the stress.

In particular, our results on the investigated biochemical mechanisms of protection of the quite drought tolerant and xeric-adapted *Q. cerris* show that its tolerance to soil water deficit might mainly rely on the control of the antioxidant defense system, while its thermotolerance can be partially explained by a high ability to modulate osmolyte production that appeared largely conferred by DMSP. In *Q. robur*, tolerance to the applied soil water deficit and high temperature conditions appears to depend to a certain extent on the highest induction of glycine betaine together with an intrinsic reserve of defensive components.

It is worth noting that in each species, the compounds present at a significant basal level in non-stress conditions are also those more responsive when stresses are applied, as for tannins and DMSP in *Q. cerris*, and for glycine betaine in *Q. robur*. This pattern suggests that the intrinsic pool of those specific metabolites that are constitutively available in each species, may entail a higher ability of these species to activate a prompt and fine-tuned defensive response when oxidative stress occurs. If these response patterns of components of the protection system indicate that the metabolism of the studied oaks is prepared for unfavorable conditions according to [47], they reveal a species-specific metabolic pre-adaptation to these environmental constraints that contributes to the stress tolerance of these oak species.

Additional experiments are needed to elucidate the function of metabolites such as DMSP and isoprenoids that have a multifunctional protective role and are interconnected with other defense mechanisms.

## Figures and Tables

**Figure 1 plants-11-01744-f001:**
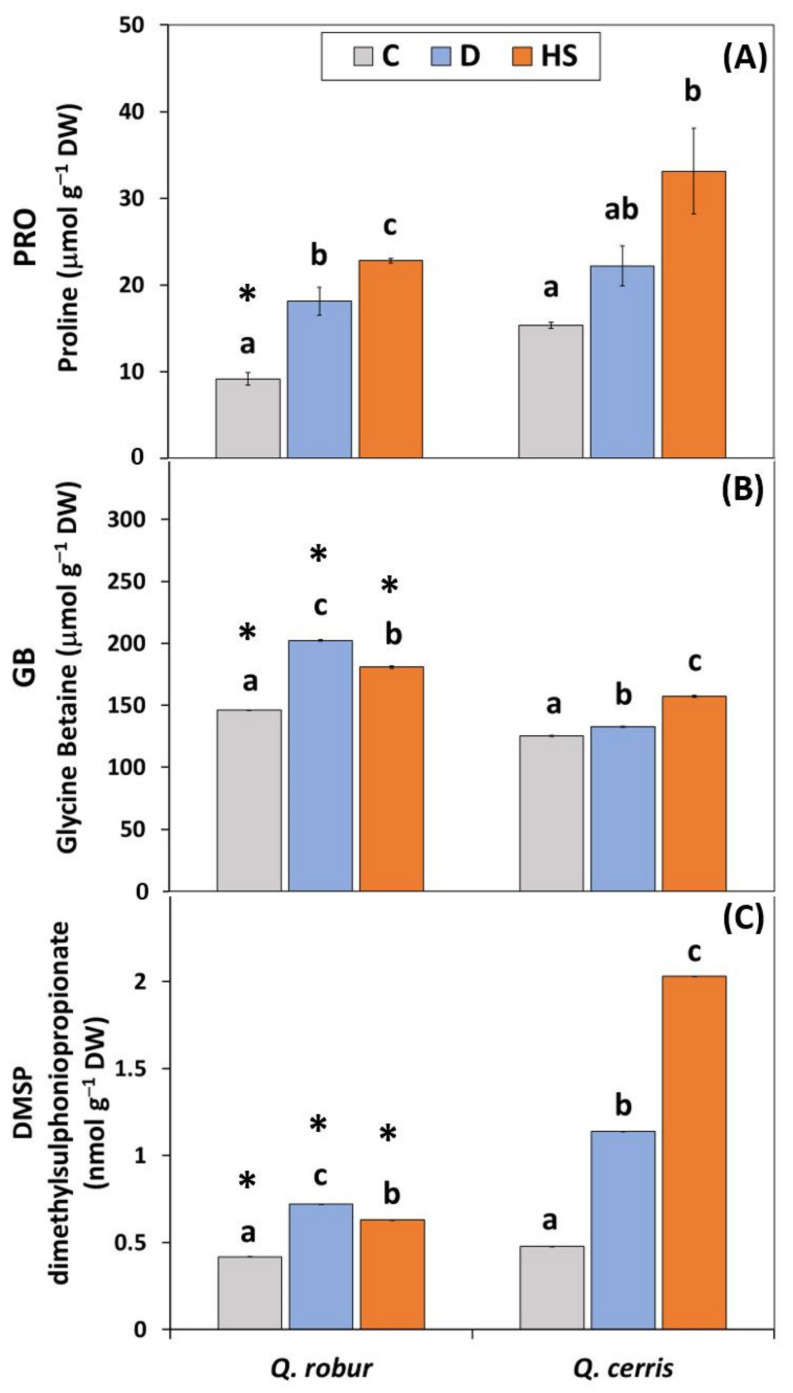
Changes in: (**A**) proline (PRO; μmol g^−1^ DW); (**B**) glycine betaine (GB; μmol g^−1^ DW); and (**C**) dimethylsulphoniopropionate (DMSP; nmol g^−1^ DW) content in leaves of *Quercus robur* and *Q. cerris*. Treatments: C: control plants under well-watered soil conditions to maintain SWC (Soil Water Content) in the range of 32–38% and daily air temperature of 25–28 °C; D: drought-stressed plants by withholding soil water for 12 days until the SWC reached values of ca. 9–11%; HS: heat-stressed plants after 6-days exposure to daily temperatures ranging between 33–47 °C. Different lowercase letters indicate significant differences among treatments within each species, while asterisks indicate significant differences between the two oak species within each treatment after one-way ANOVA with Tukey’s honestly significant difference (HSD) post hoc test (*p* ≤ 0.05). Data represent the mean ± standard deviation.

**Figure 2 plants-11-01744-f002:**
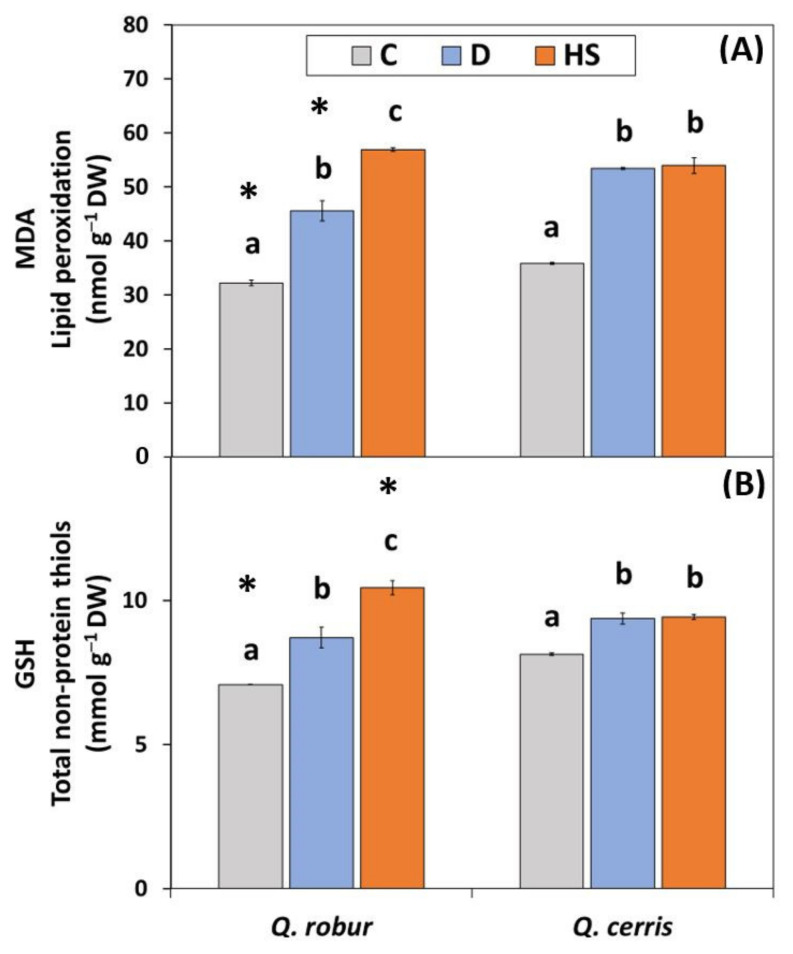
(**A**) Lipid peroxidation measured as levels of malondialdehyde (MDA; nmol g^−1^ DW) and (**B**) total non-thiol compounds measured as reduced glutathione (GSH; nmol g^−1^ DW) in leaves of *Q. robur* and *Q. cerris*. Abbreviations of the treatments, mean separation, and statistical treatments are as shown in Figure 1. Data represent the mean ± standard deviation.

**Figure 3 plants-11-01744-f003:**
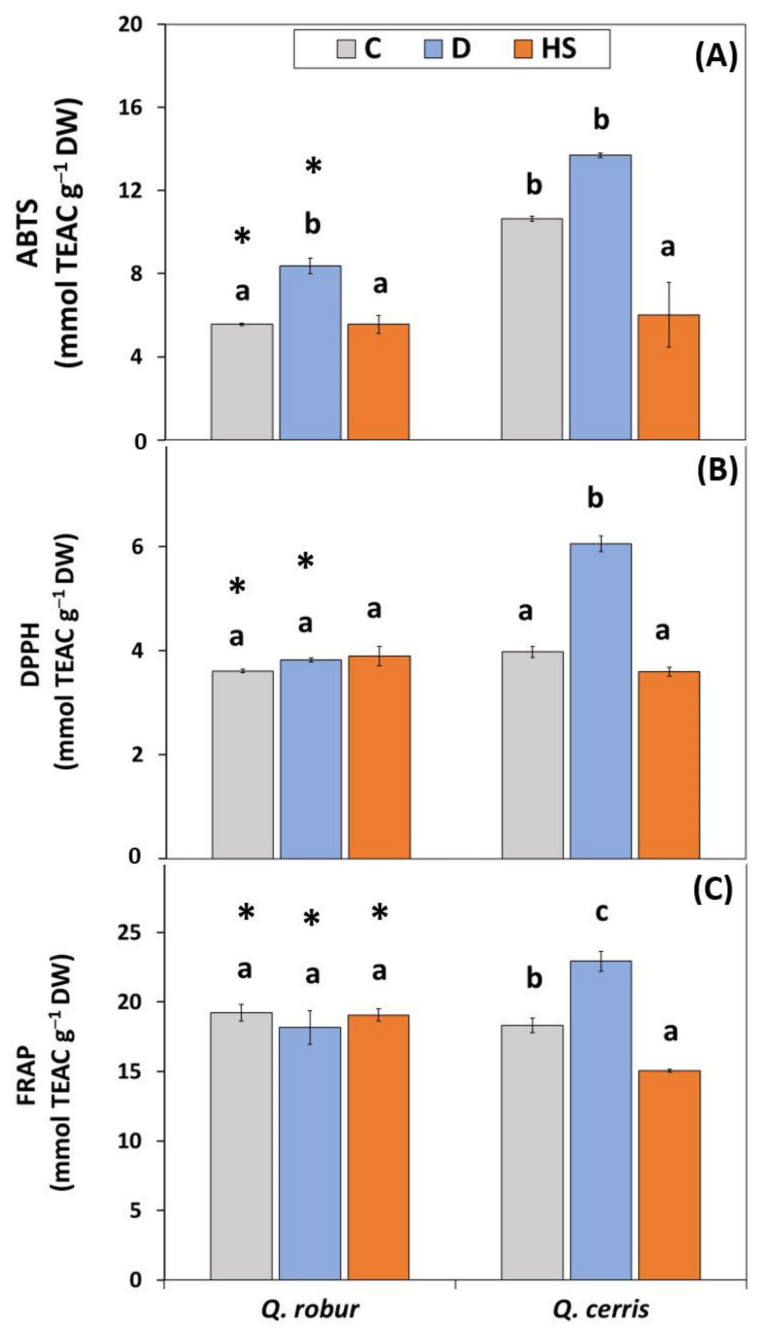
Total antioxidant capacity (mmol TEAC g^−1^ DW) measured by: (**A**) ABTS; (**B**) DPPH; and (**C**) FRAP assay in leaves of *Q. robur* and *Q. cerris*. Abbreviations of the treatments, mean separation, and statistical treatments are as shown in Figure 1. Data represent the mean ± standard deviation.

**Figure 4 plants-11-01744-f004:**
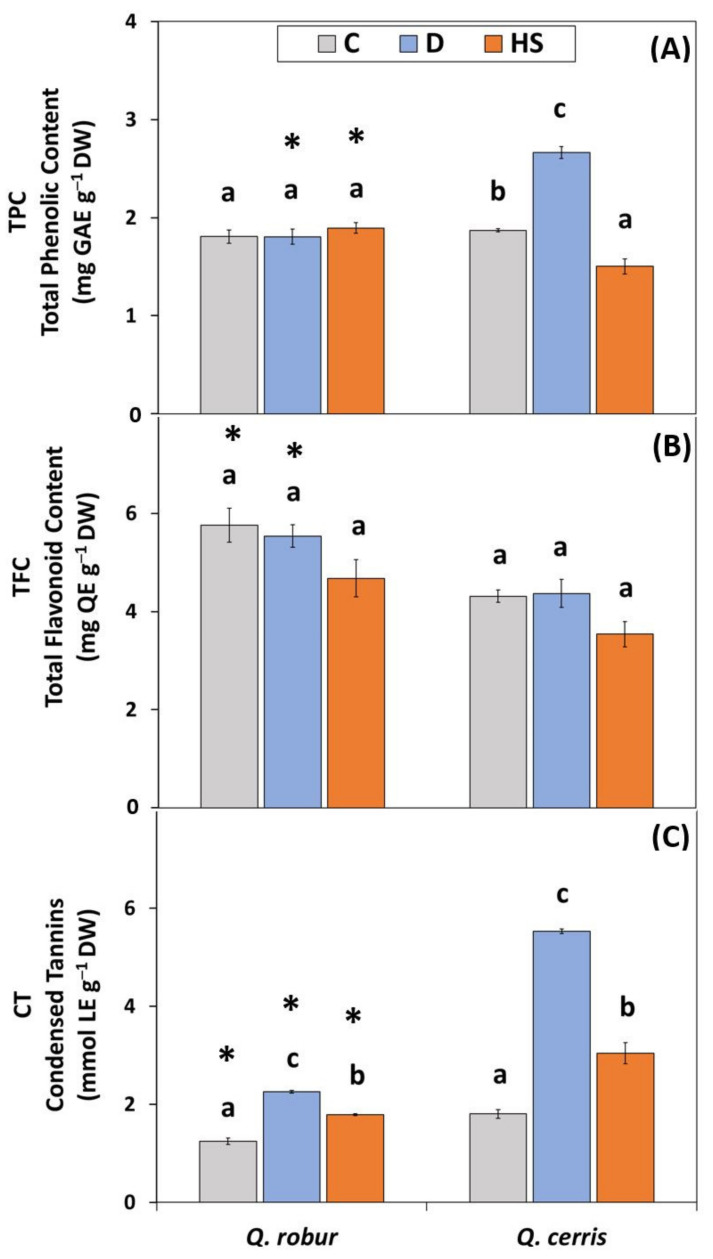
(**A**) Total phenolic content (TPC; mg GAE g^−1^ DW); (**B**) total flavonoid content (TFC; mg QE g^−1^ DW); and (**C**) condensed tannins (CT; mg LE g^−1^ DW) in leaves of *Q. robur* and *Q. cerris*. Abbreviations of the treatments, mean separation, and statistical treatments are as shown in Figure 1. Data represent the mean ± standard deviation.

**Figure 5 plants-11-01744-f005:**
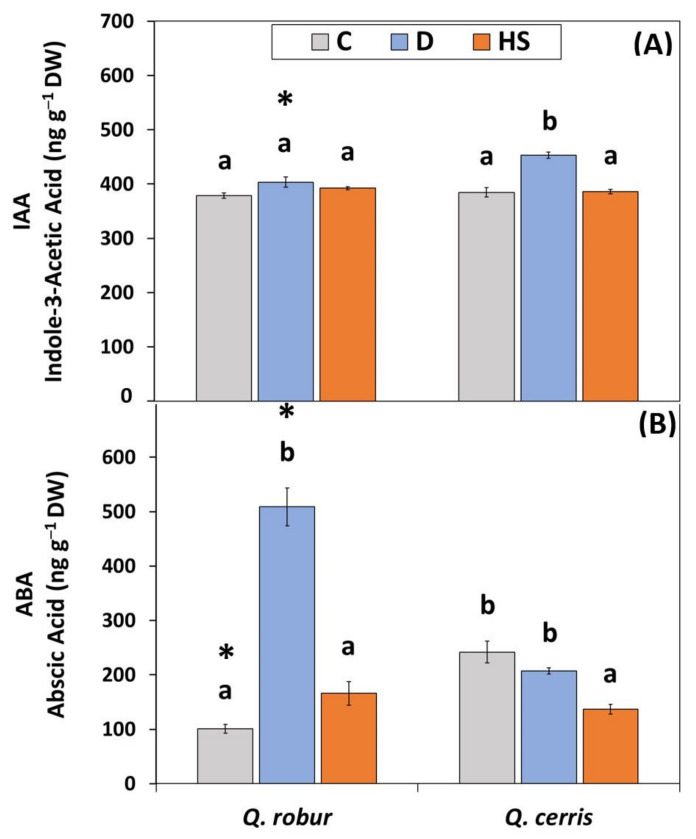
(**A**) Free indole-3-acetic acid (IAA; ng g^−1^ DW) and (**B**) abscisic acid (ABA; ng g^−1^ DW) content in leaves of *Q. robur* and *Q. cerris*. Abbreviations of the treatments, mean separation, and statistical treatments are as shown in Figure 1. Data represent the mean ± standard deviation.

**Figure 6 plants-11-01744-f006:**
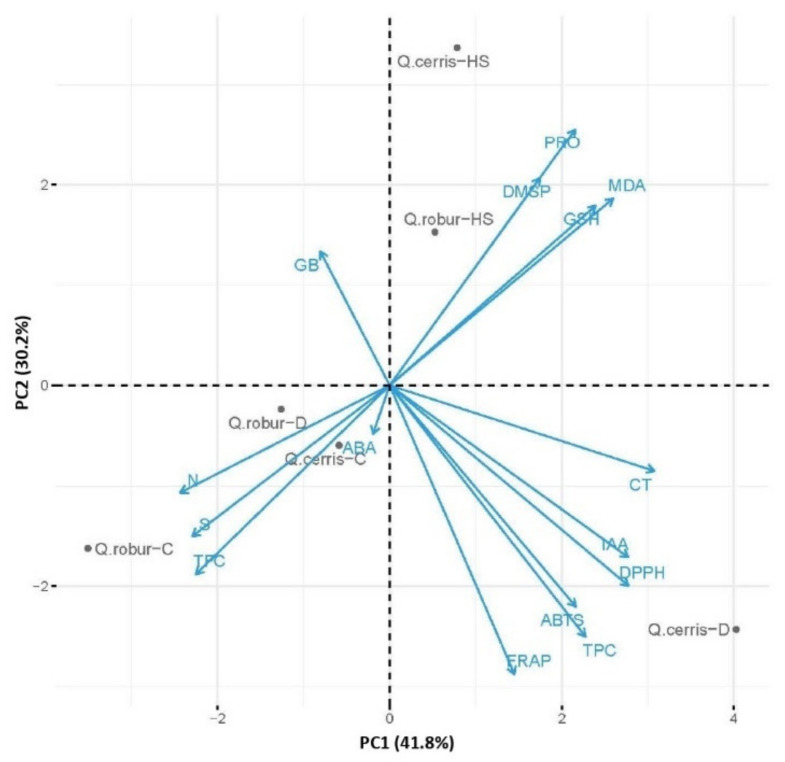
Principal component analysis (PCA) on a correlation matrix including all the biochemical parameters for control I drought-stressed (D) and high-temperatures-stressed (HS) plants of *Quercus cerris* (Q. cerris) and *Quercus robur* (Q. robur). TPC: total phenolic content; TFC: total flavonoid content; CT: condensed tannins; MDA: lipid peroxidation measured as levels of malondialdehyde; GSH: total non-thiol compounds measured as reduced glutathione; ABTS: total antioxidant capacity measured by ABTS assay; DPPH: total antioxidant capacity measured by DPPH assay; FRAP: total antioxidant capacity measured by FRAP assay; PRO: free proline; GB: glycine-betaine content; DMSP: dimethylsulphoniopropionate; IAA: free indole-3-acetic acid; ABA: abscisic acid; S: sulfur; N: nitrogen.

**Figure 7 plants-11-01744-f007:**
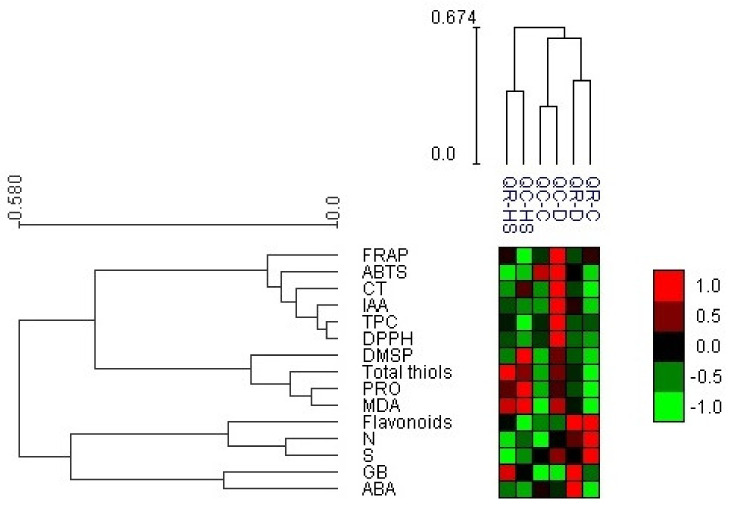
Heat map with bi-cluster analysis where red squares represent highly significant correlation of applied treatment and inspected parameter, while green squares present low interaction assessed according to corresponding Pearson’s coefficient. See legend of Figure 1 and Figure 6 for abbreviations of the treatments and parameters.

**Figure 8 plants-11-01744-f008:**
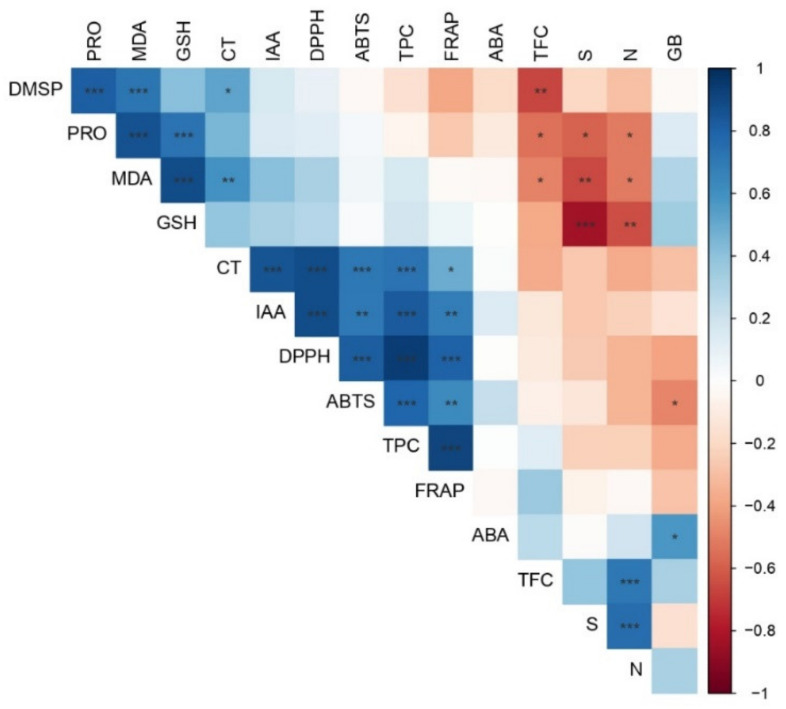
Pearson’s coefficient of correlation matrix of the examined parameters in *Q. robur* and *Q. cerris* leaves *. See legend of Figure 6 for abbreviations of parameters. The strength of the correlation is indicated by color saturation. Significance values are indicated as: * *p* < 0.05; ** *p* < 0.01; *** *p* < 0.001.

**Figure 9 plants-11-01744-f009:**
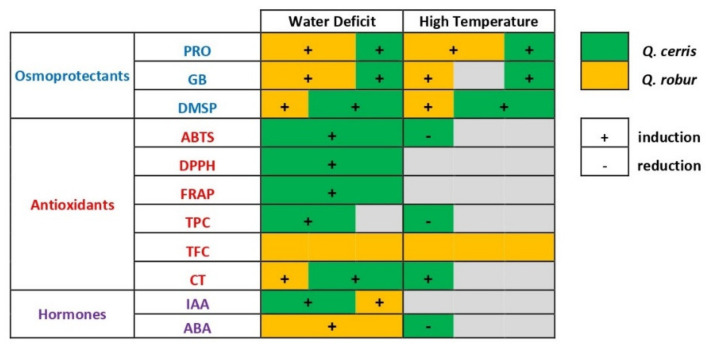
Schematic representation of the soil-water-deficit and high-air-temperatures-induced changes in leaf level content of osmoprotectants, antioxidants, and hormones in *Q. cerris* and *Q. robur*. The size of the boxes reflects the relative induction or reduction of one species compared with the other. See legend of Figure 6 for abbreviations of the parameters.

**Table 1 plants-11-01744-t001:** Responses to different biochemical components of the plant defense system to single drought or high temperature stress in young trees of *Q. robur* and *Q. cerris*.

Plant Species	Plant Age	Growth Conditions	Stress Design	Response to Drought	Response to High Temperature	Literature
*Q. robur*	10 weeks	Potted plants under controlled conditions	Drought: withholding water for about 21 days	↓ Redox ratio of ascorbate and glutathione ↑ MDA ↓ Carotenoids		Schwanz and Polle, 2001
*Q. robur*	5 years	Potted plants in greenhouse	Drought: withholding water to reach soil moisture level of 13%	↑ Carbohydrates (hexoses) and polyols ↑ Proline and Proline derivatives ‖ Glycine betaine		Spieß et al., 2012
*Q. robur*	3–5 years	Potted plants under controlled conditions	Drying-rewetting cycles (reduced irrigation for about 20 and 30 days) High temperature (+1/2 °C compared to controlled conditions)	↑ Proline, GABA ↑ Glutathione ‖ Total and Reduced Ascorbate ‖ Cysteine and γ-glutamyl cysteine	‖ Proline, GABA ‖ Glutathione ‖ Total and Reduced Ascorbate ‖ Cysteine and γ-glutamyl cysteine	Hu et al., 2013
*Q. robur*	3–5 years	Potted plants under controlled conditions	Drying-rewetting cycles (reduced irrigation for about 20 and 30 days) High temperature (+1/2 °C compared to controlled conditions)	↑ Total and specific amino acids-N and Glutamine	‖ Total and specific amino acids-N	Hu et al., 2015
*Q. cerris*	3 years	Potted plants in greenhouse under controlled conditions	Drought (daily irrigation with 30% effective evapotranspiration for 2 weeks)	↑ Proline, MDA ↑ Chlorophylls ‖ Carotenoids (xantophylls)		Cotrozzi et al., 2015
*Q. cerris*	3 years	Potted plants in greenhouse under controlled conditions	Drought (daily irrigation with 20% effective evapotranspiration for 2 weeks)	↑ Proline, MDA ↑ Carotenoids, chlorophylls, β-carotene, α-tocopherol, ABA ‖ Carbohydrates (hexoses)		Cotrozzi et al., 2016

↑: increased; ↓: decreased; ‖: unchanged.

## Data Availability

Not applicable.

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
