# Peer review of "Species-Level Differences in Osmoprotectants and Antioxidants Contribute to Stress Tolerance of Quercus robur L., and Q. cerris L. Seedlings under Water Deficit and High Temperatures"

_plants, 2022, doi:10.3390/plants11131744_

Round 1

Reviewer 1 Report

Line 133-140-long sentence, which should be splited. 

The hypothesis of the current work should be added in the section of introduction.

The information in Table 1 can be removed, which can be mentioned in details in the section of introduction. The remarks of A, B, C in each figure should not be red- could be black or grey or. 

The font size is too small to follow in the figures. Check the ANOVA of figure 4a since there is no b, but a, b, c remarked above bar.

The author mentioned there were one way and two way ANOVA. So what about the significant letter shown in the figures? Did it take the species effect into account or not? The information should be provided in the caption of figures.

The conclusion is too long, where the important information should be extracted. The figure 9 should be moved to discussion instead of conclusion. More importantly, how the author connect the results from warming and drought? Is there any connection or significantly different responsive process? 

Reviewer 2 Report

The manuscript plants-1772348 has been properly presented in an easy understanding way for the readers. The experimental design was reasonable, and the methods used were the standard ones.

I- Abstract section

- Line 25 correct Q. cerris and Q. robur to Q. cerris (Q. cerris) and Quercus robur (Q. robur)

2- Introduction is OK

3- Results section is OK

4- Discussion section is OK

5- M&M section

- Line 588: Why 90 days?

Conclusion section is OK

Figures and Tables are OK

Reviewer 3 Report

 General comments

I have read the manuscript (plants-172348). Entitle: Species-Level Differences in Osmoprotectants and Antioxidants Levels Contribute to Stress Tolerance of Quercus robur L., and Q. cerris L. Seedlings under Water Deficit and High Temperatures written by Marko Kebert et. al., for publication of plants MDPI. In this study, the author investigates the changes in hormones. Between two species of seedlings and differently increasing the levels of Osmo protectants and antioxidants based on the stress and their treading-off. Author mainly found that the Q. cerris had a higher antioxidant capacity compared to Q. robur, which showed a lower investment in the antioxidant system. The exposure to high temperatures induced a strong osmoregulation capacity in both species.

Our results were discussed in 35 terms of pre-adaptation and stress-induced metabolic patterns as related to species-specific stress 36 tolerance features. The overall research is well conducted, and research is obvious application potential for the readers because this research well-discuss the terms of pre-adaptation and stress-induced metabolic patterns as related to species-specific stress tolerance features. In this sense, the manuscript is much valuable. However, I found some points, especially the flow of the text and lack of potential references, and lack of connection of story in different paragraphs, especially in the introduction and discussion sections. The author should provide enough examples and their interpretation of different traits of physiological and biochemical responses by the latest and appropriate references, some of which I mentioned below. Overall after I evaluate and request the author for this manuscript as a “MAJOR REVISION”.

 Major suggestions

 1)  Introduction: The introduction is well-starting climate change issues that affects the plant vitality, health status and distribution but author should correct many think in this section. Author should mention the main theme of antioxidant and secondary metabolites under stress conditions (light/metal/drought/flood) and release the ROS (why ROS is emerging in stress conditions?). Refer to these two articles for better clarify (1) https://doi.org/10.1038/s41598-019-55889 (2) https://doi.org/10.1016/j.scitotenv.2021.146466 and mention somewhere in that paragraph “abiotic stress especially environmental stress (e.g. UV radiation) plant produces the ROS when the plant exposed to the stress condition and plant produce antioxidant, flavonoids, and secondary metabolites play to the role for protecting the plant for detoxifying ROS and protect the plant to protect the abnormal condition (i.e. stress) and protein and amino acid stabilization”.

2) Hypothesis and objectives of the study: Author shoulis is is d rephrase the text more deeply in the last two paragraph of the introduction specially Ln. 130-145 by focusing on the hypothesis of the study and also connecting the objectives of the study. Author tries to mention but this is still not clear. The hypothesis of the study is an important thing, and it gives another strength to the introduction. The hypothesis should be very clear in the introduction sections because, without appropriate literature, questions, or hypotheses in the introduction section the entire text will be unclear. The author should give special attention and the sequential presentation of the content in the introduction with presenting the hypothesis of the study.

3) Discussion: Discussion is well written, but it should be concise.  Author much improved the subsection 3.4 “drought impact differently the ABA hormonal balance” by referring latest drought-related articles and focus these statements. Generally, when plants are exposed to the drought, the level of ABA is increased, moreover, the higher the drought intensity higher the ABA concentration. Which is provide the signal from the soil to the root and plant system thoroughly. The ABA hormone regulation is one of the important pathways. The text related to the ABA under different drought intensities is well described in this article https://DOI:10.1016/j.scienta.2018.11.021.

Some other suggestions

 4)  Line no. 41: The research is mainly related to the drought and heat but but most of the introduction is covered by ROS, hormonal and secondary metabolites, and antioxidants, these are also need but the primary drought effect come first. Therefore, author should focus on equality the primary effect of drought. Therefore, read and mentioned this as a reference. Entitle: Entitle “Response of drought stress in prunus sargentii and larix kaempferii ...https://doi.org/10.1016/j.foreco.2020.118099” “drought reduced the morphological and physiological traits, reduce the leaf water potential and sap movement due to alternation of xylem anatomical features in the plants”. Then only the author should descript the other’s abiotic stress.

5) Line no. 318: Why author analyze the heat map by bi-cluster analysis? It may reason due to cluster distance of the oak species.?

6) Line no. 607: Author mentions the methodological section has too much detail. Actually, scientific paper to detail describes the method sections is not scientific and a instead, we should refer the someone’s paper who previously describe this method. The author should follow this in section 4.2 in MM and accordingly follow others.

7) Line no. 638: Author mentions the methodological section has too much detail. Similar to above author should follow the above rule in other sub-section of methodology such as 4.3 and 4.4 and 4.5 as well.

8) Line no.753 (Conclusion): The conclusion for me comes off as repetitive of the abstract or a summary of the results section. I would love to read striking points and take-home messages that will linger in the readers’ minds. What is the novelty, how does the study elucidate some questions in this field, and the contributions the paper may offer to the scientific community?

9) Line no. 836 (Reference): please double-check the citations, their style, spell check, and other grammatical errors. moreover, I request to the authors for revision throughout the manuscript according to the journal rules.

 Good Luck!

Round 2

Reviewer 3 Report

Dear Author

I have read the revised manuscript (plants -1772348). Titled: Species-Level Differences in Osmoprotectants and Antioxidants Levels Contribute to Stress Tolerance of Quercus robur L., and Q. cerris L. Seedlings under Water Deficit and High Temperatures for publication in plants MDPI. This is the second submission made by the author. The author addressed all the questions and suggestions that I raised the issue in the review of the original manuscript. I satisfy the author’s revisions throughout the paper. Author well addresses the abstract issues. Especially author improved the introduction and discussion section very well inflow. Now, this manuscript improved the flow of writing, which was comparatively shallow in the original version but in this revised copy author addressed all the quarries and suggestions very well. Before accepting this manuscript if there is anything needed to be revised by the author, especially English grammar, or spell check, I request this manuscript is currently in “Minor Revision” and any grammatical error author may improve in this stage. Thank you.